# The Relationship between ECOG-PS, mGPS, BMI/WL Grade and Body Composition and Physical Function in Patients with Advanced Cancer

**DOI:** 10.3390/cancers12051187

**Published:** 2020-05-08

**Authors:** Ross D. Dolan, Louise E. Daly, Claribel Pl. Simmons, Aoife M. Ryan, Wei Mj. Sim, Marie Fallon, Derek G. Power, Andrew Wilcock, Matthew Maddocks, Michael I. Bennett, Caroline Usborne, Barry J. Laird, Donald C. McMillan

**Affiliations:** 1Academic Unit of Surgery, School of Medicine, University of Glasgow, Glasgow Royal Infirmary, Glasgow G31 2ER, UK; 2115962s@student.gla.ac.uk (W.M.S.); Donald.McMillan@glasgow.ac.uk (D.C.M.); 2School of Food and Nutritional Sciences, College of Science, Engineering and Food Science, University College Cork, Cork, Ireland; louisedaly@umail.ucc.ie (L.E.D.); a.ryan@ucc.ie (A.M.R.); drdgpower@gmail.com (D.G.P.); 3Edinburgh Cancer Research Centre, Institute of Genetics and Molecular Medicine, University of Edinburgh, Edinburgh EH4 2XR, UK; claribel@doctors.org.uk (C.P.S.); marie.fallon@ed.ac.uk (M.F.); barry.laird@ed.ac.uk (B.J.L.); 4University of Nottingham, Hayward House, Nottingham City Hospital, Hucknall Road, Nottingham NG5 1PB, UK; andrew.wilcock@nottingham.ac.uk; 5Cicely Saunders Institute of Palliative Care Policy & Rehabilitation, King’s College London, Bessemer Road, London SE5 9PJ, UK; matthew.maddocks@kcl.ac.uk; 6Academic Unit of Palliative Care, Leeds Institute of Health Sciences, School of Medicine, University of Leeds, Room 10.37, Worsley Building, Clarendon Way, Leeds LS2 9LU, UK; M.I.Bennett@leeds.ac.uk; 7Glan Clwyd Hospital, Bodelwyddan, Denbighshire LL18 5UJ, UK; caroline.usborne@wales.nhs.uk

**Keywords:** advanced cancer, systemic inflammation, Glasgow prognostic score, body composition, ECOG, physical function testing, computed tomography

## Abstract

Cancer remains one of the leading causes of mortality worldwide and the associated reduction in physical function has a marked impact on both quality of life and survival. The aim of the present study was to examine the relationship between Eastern Cooperative Oncology Group-Performance status (ECOG-PS), modified Glasgow Prognostic Score (mGPS), Body Mass Index/Weight Loss grade (BMI/WL grade), and Computerised Tomography (CT)-derived body composition measurement and physical function in patients with advanced cancer. Nine sites contributed prospective data on patient demographics, ECOG-PS, mGPS, physical function tests, and CT-derived body composition. Categorical variables were analysed using χ^2^ test for linear-by-linear association, or χ^2^ test for 2-by-2 tables. Associations were analysed using binary logistic regression. A total of 523 cancer patients (266 males, 257 females) were included in the final analysis and most had metastatic disease (83.2%). The median overall survival was 5.6 months. On multivariate binary logistic regression analysis, a high ECOG-PS remained independently associated with a low skeletal muscle index (*p* < 0.001), low skeletal muscle density (*p* < 0.05), and timed up and go test failure (*p* < 0.001). A high mGPS remained independently associated with a low skeletal muscle density (*p* < 0.05) and hand grip strength test failure (*p* < 0.01). A high BMI/WL grade remained independently associated with a low subcutaneous fat index (*p* < 0.05), low visceral obesity (*p* < 0.01), and low skeletal muscle density (*p* < 0.05). In conclusion, a high ECOG-PS and a high mGPS as outlined in the ECOG-PS/mGPS framework were consistently associated with poorer body composition and physical function in patients with advanced cancer.

## 1. Introduction

Cancer remains one of the leading causes of mortality worldwide and is responsible for 8.8 million deaths each year. In westernised countries, it has been estimated that one in three people will develop cancer in their lifetime and one in four will die from it [1,2].

The importance of cachexia syndrome, with escalating nutritional and functional decline leading to poor clinical outcomes, is well recognised [3]. However, how this complex syndrome is best defined is the subject of continuing debate. Clearly, defining any syndrome is difficult due to its multifaceted nature. However, in such circumstances one may resort to the duck test approach: “If it looks like a duck, swims like a duck, quacks like a duck, then it probably is a duck”. Such abductive reasoning has been commonly used to settle legal cases and more recently has gained popularity in artificial intelligence.

In the context of cancer cachexia, a number of factors have been shown to impact independently on quality of life (including functional and symptom scores) and survival. These include Eastern Cooperative Oncology Group-Performance status (ECOG-PS) and the systemic inflammatory response (modified Glasgow Prognostic Score, mGPS), both of which have been extensively validated [4,5,6,7]. More recently, based on an international consensus, body mass index/weight loss (BMI/WL) grades have been shown to impact independently on quality of life and survival [8,9,10].

Recently, these three criteria for the definition of cancer cachexia were directly compared and all three independently predicted survival in patients with advanced cancer [11]. However, BMI/WL grade was low risk in approximately 50% of patients and ECOG-PS and mGPS were independently associated with survival in this group. Therefore, to further investigate the clinical utility of these three criteria to define cachexia, the aim of the present study was to examine the relationship between ECOG-PS, mGPS, BMI/WL grade, and Computerised Tomography (CT)-derived body composition and physical function tests in patients with advanced cancer.

## 2. Patients and Methods

### 2.1. Patients

A biobank of data from patients with advanced cancer was analysed. All data were collected prospectively across 9 sites in the UK and Ireland (cancer centres, hospitals, and specialist palliative care units) over a five-year period (2011–2016) [9,11,12]. Eligible patients provided written informed consent, were adults, had advanced cancer including all cancer subtypes (defined as metastatic cancer with histological, cytological or radiological evidence, that was locally advanced, or receiving anti-cancer therapy with palliative intent) and had the ability to comply with study procedures including provision of a venous blood sample (taken on the day of consent). Patients were either inpatients or outpatients, undergoing anti-cancer therapy with a palliative intent including best supportive care. The study had ethical approval in both the UK and Ireland (West of Scotland Ethics Committee UK: 18/WS/0001 (18/01/2018) and Cork Research Ethics Committee Ireland: ECM 4 (g) (03/03/2015)) and was conducted in accordance with the Declaration of Helsinki. Furthermore, the study conformed to the STROBE guidelines for cohort studies [13].

#### 2.1.1. Prognostic Markers

Patient’s age, sex, and clinicopathological characteristics were recorded within 3 months prior to study entry. Prognostic tools/factors validated in a recent systematic review by Simmons and co-workers were used in the analysis [14].

Patients were categorized according to their ECOG-PS into five district grades (grade 0–4) as previously described [15]. The mGPS was constructed as previously described (CRP ≤ 10 mg/L = 0, CRP > 10 mg/L & albumin ≥ 35 g/L = 1, CRP > 10 mg/L and albumin < 35 g/L = 2) [16,17]. An autoanalyzer was used to measure serum CRP (mg/L) and albumin (g/L) concentrations (Architect; Abbot Diagnostics, Maidenhead, UK). Patients were categorized according to the BMI-adjusted weight loss grade into one of five distinct weight loss grades (grades 0–4) as previously described [8,9].

#### 2.1.2. Body Composition

CT images were obtained at the level of the third lumbar vertebra [18]. Patients whose scans were taken ≥ 3 months prior to study entry, who had significant movement artefact, or who were missing the region of interest were excluded. CT images were analysed using NIH Image J version 1.47 (U.S. National Institutes of Health, Bethesda, USA) or OsiriX software version 4.1.1 (OsiriX, Geneva, Switzerland). Both imaging software packages have been shown to provide excellent agreement for body composition measures [19]. Region of interest (ROI) measurements were made of visceral fat areas, subcutaneous fat areas (Table 1), and skeletal muscle areas (cm^2^) (Table 1) using standard Hounsfield Unit (HU) ranges (adipose tissue -190 to -30, and skeletal muscle -29 to +150). These were then normalised for height^2^ to create indices: total fat index (cm^2^/m^2^), subcutaneous fat index (cm^2^/m^2^), visceral fat index (cm^2^/m^2^), and skeletal muscle index (cm^2^/m^2^). Skeletal muscle radiodensity (HU) was measured from the same ROI used to calculate skeletal muscle index, as its mean HU.

Visceral obesity was defined by Doyle and colleagues as a visceral fat area >160cm^2^ for male patients and >80cm^2^ for female patients [23]. High subcutaneous fat was defined by Ebadi and colleagues as a subcutaneous fat index ≥ 50.0 cm^2^m^2^ in males and ≥42.0 cm^2^m^2^ in females [20]. Low skeletal muscle index was defined as described by Martin and colleagues, with a skeletal muscle index <43 cm^2^/m^2^ if BMI <25 kg/m^2^ and skeletal muscle index <53 cm^2^/m^2^ if BMI ≥25 kg/m^2^ in male patients and skeletal muscle index <41cm^2^/m^2^ in female patients [22]. Low skeletal muscle radiodensity was defined by Martin and colleagues as an skeletal muscle radiodensity <41HU in patients with BMI <25 kg/m^2^ and <33HU in patients with BMI ≥25 kg/m^2^ (Table 1) [22].

Two individuals performed scan measurements (Dolan and Daly). In order to assess accuracy, inter-rater reliability was measured in a test cohort of 20 patient images. Inter-class correlation coefficients were 0.986 for skeletal muscle area and 0.964 for skeletal muscle radiodensity. Investigators were blind to patient’s demographic and clinico-pathological status.

#### 2.1.3. Physical Function

Eastern Cooperative Performance Status (ECOG-PS), timed up and go, two-minute walk, and hand grip strength tests, as well as the presence of metastases and weight loss over the preceding three months to study entry, were assessed by either the treating clinician or clinical research staff. Timed up and go test and two-minute walk test completion were recorded contemporaneously with completion being recorded as a test pass. A failure of timed up and go has previously been defined by Kear and co-workers for patients under 60 and by Rockwood and co-workers in patients over 60 [24,25,26]. A failure of the two-minute walk test has previously been defined by Bohannon and co-workers for male and female patients between 18 and 85 years of age [27]. A weak hand grip strength test was defined by Studenski and co-workers as <26 kg in men and <16 kg in women [28]. Patients who achieved a hand grip strength results below the above thresholds were deemed to have failed the hand grip strength test.

### 2.2. Statistical Analysis

Body composition measurements were presented as median and range and compared using Mann–Whitney or Kruskal–Wallis tests. Categorical variables were analysed using χ^2^ test for linear-by-linear association, or χ^2^ test for 2-by-2 tables.

Associations between ECOG-PS, mGPS, BMI-WL grades, body composition, physical function tests, and survival were analysed using univariate and a multivariate backward conditional approach. A *p* < 0.05 was applied to inclusion at each step in the multivariate analysis.

Missing data were excluded from analysis on a variable by variable basis. Two-tailed *p* values <0.05 were considered statistically significant. Statistical analysis was performed using SPSS software version 21.0. (SPSS Inc., Chicago, IL, USA).

## 3. Results

A total of 523 patients (266 males, 257 females) satisfied the inclusion criteria. The relationship between clinicopathological characteristics, body composition, and physical function is shown in Table 2. The majority of patients were over 65 (56.8%), had a BMI >25 kg/m2 (50.1%), and had metastasis (83.2%). Gastrointestinal (34.4%) and lung (31.7%) cancers were the most common tumours. The median overall survival was 5.6 months (95% CI: 5.1–6.0 months). At the date of censoring, 318 patients (61%) were dead. Median follow-up time for patients that had died was 10.5 months (95% CI: 9.0–12.1 months).

The relationship between ECOG-PS and measures of body composition and physical function are shown in Table 3. ECOG-PS was significantly associated with skeletal muscle index (*p* < 0.05), skeletal muscle radiodensity (*p* < 0.001) and timed up and go (*p* < 0.001).

The relationship between mGPS and measures of body composition and physical function are shown in Table 4. mGPS was significantly associated with skeletal muscle radiodensity (*p* < 0.01), timed up and go test failure (*p* ≤ 0.001), and hand grip strength test failure (*p* < 0.01).

The relationship between BMI/WL grade and measures of body composition and physical function are shown in Table 5. BMI/WL grade was significantly associated with visceral obesity (*p* < 0.05) and skeletal muscle radiodensity (*p* < 0.01).

Low skeletal muscle radiodensity was significantly associated with timed up and go test failure (n = 192, *p* = 0.015) and hand grip strength failure (n = 100, *p* = 0.042).

The relationship between ECOG-PS, mGPS, BMI/WL grade, and subcutaneous fat index in patients with advanced cancer is shown in Table 6. On multivariate binary logistic regression analysis, BMI/WL grade (OR 0.62, 95%CI 0.40–0.97, *p* < 0.05) remained independently associated with a high subcutaneous fat index.

The relationship between ECOG-PS, mGPS, BMI/WL grade and high visceral obesity is shown in Table 6. On multivariate binary logistic regression analysis, BMI/WL grade (OR 0.57, 95%CI 0.38–0.87, *p* < 0.01) remained independently associated with a high visceral obesity.

The relationship between ECOG-PS, mGPS, BMI/WL grade and low skeletal muscle index is shown in Table 6. On multivariate binary logistic regression analysis, ECOG-PS (OR 1.90, 95%CI 1.51–2.39, *p* < 0.001) remained independently associated with a low skeletal muscle index.

The relationship between ECOG-PS, mGPS, BMI/WL grade and low skeletal muscle radiodensity is shown in Table 6. On multivariate binary logistic regression analysis, ECOG-PS (OR 1.68, 95%CI 1.11–2.55, *p* < 0.05), mGPS (OR 1.32, 95%CI 1.01–1.73, *p* < 0.05) and BMI/WL grade (OR 1.50, 95%CI 1.02–2.19, *p* < 0.05) remained independently associated with a low skeletal muscle radiodensity.

The relationship between ECOG-PS, mGPS, BMI/WL grade and timed up and go test is shown in Table 6. On multivariate binary logistic regression analysis, ECOG-PS (OR 5.84, 95%CI 3.79–9.00, *p* < 0.001) remained independently associated with timed up and go failure.

The relationship between ECOG-PS, mGPS, BMI/WL grade and hand grip strength is shown in Table 6. On multivariate binary logistic regression analysis, mGPS (OR 1.95, 95%CI 1.29–2.97, *p* < 0.01) remained independently associated with hand grip strength failure.

## 4. Discussion

Over the last decade or so there has been increasing interest in identifying objective criteria to define cancer cachexia. This has proven problematic since cancer cachexia is a syndrome impacting on quality of life, body composition, physical function, and survival. In the present study, candidate criteria were directly compared in terms of their relationship with measures of body composition and physical function. ECOG-PS and mGPS were consistently associated with low skeletal muscle mass and function and therefore, together with our previous study [11], both ECOG-PS and mGPS would appear to pass the duck test as criteria to define cancer cachexia.

In the present study, poor performance status was significantly associated with low skeletal muscle index, low skeletal muscle radiodensity, and timed up and go test failure but not hand grip strength test failure. Furthermore, high mGPS was significantly associated with low skeletal muscle radiodensity, timed up and go test failure, and hand grip strength test fail. In contrast, high BMI/WL grade was significantly associated with high subcutaneous fat index, high visceral obesity, and low skeletal muscle radiodensity. Therefore, BMI/WL grade appears to capture elements of the decline in fat mass. The present and previous [11] results clearly need to be repeated to prove the clinical utility of the ECOG-PS/mGPS framework. However, if this proves to be the case (and these observations are readily repeated) there are a number of important implications for the future diagnosis and treatment of cancer cachexia. The present results would suggest that in addition to ECOG-PS, mGPS is useful in defining the syndrome of cancer cachexia. Therefore, the ECOG/mGPS framework should be considered as part of routine assessment prior to treatment in patients with advanced cancer.

In the present study it was of interest that low skeletal muscle radiodensity was significantly associated with timed up and go test failure (*p* < 0.05) and hand grip strength test failure (*p* < 0.05). These results would be consistent with the results of a recent study by Williams and co-workers who reported that skeletal muscle radiodensity was related to physical function impairments including activities of daily living (ADL), climbing stairs, walking, and timed up and go [29]. Furthermore, the presence of systemic inflammatory response degrades the quality of the skeletal muscle [30]. If this were to be the case then it might be anticipated that downregulation of the systemic inflammatory response, compared with placebo, would result in better preservation of muscle density, muscle strength, and performance status. This hypothesis is the subject of a number of ongoing randomised clinical trials. For example, there is a randomised placebo controlled phase III trial underway of a multimodal intervention (exercise, nutrition, anti-inflammatory medication) in patients with advanced lung or pancreatic cancer undergoing anti-cancer therapy with palliative intent (NCT02330926) [31]. The aim of this trial is to prevent or attenuate loss of weight, muscle, and physical function using a multimodal intervention which is anti-inflammatory. The findings from the associated phase II trial provide grounds for optimism for the ongoing phase III trial [32].

In the present study, three criteria were considered to define cachexia. With reference to ECOG-PS this has long been considered a cornerstone of assessment by oncologists and palliative physicians. With the increasing integration of oncology and palliative care this is likely to remain an important part of the assessment of the patient with advanced cancer. It may be that other objective measurements of “real life” performance status will more consistently reflect ECOG-PS, such as activity trackers (e.g., Fitbit) [33,34,35]. With reference to the mGPS there has been in recent years extensive validation of its use in patients with advanced care, and routine assessment is now advocated [36,37]. Of the present criteria considered, it is the only one that is completely objective as it relies on two routine, laboratory-derived values. Indeed, it has been termed “laboratory cachexia” as its values become increasingly abnormal towards death [38] and the mGPS above has been used to define cancer cachexia [39,40]. There are other measures of the systemic inflammatory response that have been shown to have prognostic value, such as the neutrophil lymphocyte ratio which can be collected as part of the routine differential white cell count. However, such ratios have not been well defined and their relationship with the syndrome of cachexia has not been shown [7,41,42]. With reference to BMI/WL grade it is not clear whether this has additional value to other nutritional risk screening tools such as the Malnutrition Universal Screening Tool (MUST) and the Patient Generated Subjective Global Assessment (PG-SGA) that are in routine clinical use [36,40]. Therefore, further comparative studies are required to establish the value of BMI/WL grade as a measure of cachexia in patients with advanced cancer.

The findings of the present study may also help inform regulatory endpoints in the arena of trials treating cancer cachexia. To date there has been a lack of concordance in regulatory guidance between the EMA and FDA regarding endpoints [43] whilst previously agreed endpoints of skeletal muscle mass and function have not been realized in multiple clinical trials of varying agents [44,45,46,47]. It may be that moderating the systemic inflammatory response in patients with advanced cancer will produce more reproducible gains.

Limitations of the present study include that body composition measures and physical function test data were not available in all patients. In the present study the data were analysed according to clinically relevant criteria (previously reported to be associated with clinical outcomes such as survival) rather than statistical criteria. Specifically, categorical rather than continuous analysis was used and since only 10 out of 403 subjects passed the two-minute walk test this was excluded from further analysis. Furthermore, the cohort was relatively heterogenous with different cancer types and specific stages of disease. However, when further stratification of the results was carried out for both lung cancers and gastrointestinal cancers in particular (n ~180 each, Appendix A), similar results were obtained on univariate and multivariate analysis to that of the combined cohort, suggesting that the relationships between ECOG-PS, mGPS, BMI/WL grade and skeletal muscle index, skeletal muscle density and physical function were not specific to cancer type. With reference to stage of disease, more than 80% of patients had metastatic disease on study entry and therefore the heterogeneity of this cohort would have been unlikely to confound the present results. Importantly, the present results are likely to represent the type of patient cohort being treated by both oncologist and palliative care physicians. Further work is required to define these relationships in specific tumour types and at specific stages of disease. Furthermore, objective ongoing measurements of physical function such as the use of Fitbit monitors would be of considerable interest.

## 5. Conclusions

In summary, a high ECOG-PS and a high mGPS as outlined in the ECOG-PS/mGPS framework were consistently associated with poorer body composition and physical function in patients with advanced cancer. The simplicity and clinical utility of this framework mean that it can be readily incorporated into the routine assessment of patients with advanced cancer.

## Figures and Tables

**Table 1 cancers-12-01187-t001:** CT-derived body composition measures and thresholds used.

Body Composition Measurement
**High subcutaneous fat index [20]:**
Subcutaneous fat area: Males >50.0 cm^2^m^2^ and Females >42.0 cm^2^m^2^
**Visceral obesity [21,22]:**
Visceral fat area: Males >160 cm^2^ and Females >80 cm^2^
**Sarcopenia**
**Low skeletal muscle index [22]:**
Males: BMI <25 kg/m^2^ and skeletal muscle index <43 cm^2^m^2^ or BMI >25 kg/m^2^ and skeletal muscle index <53 cm^2^m^2^ Females: BMI <25 kg/m^2^ and skeletal muscle index <41 cm^2^m^2^ or BMI >25 kg/m^2^ and skeletal muscle index <41 cm^2^m^2^
**Myosteatosis**
**Low skeletal muscle radiodensity [22]:**
BMI <25 kg/m^2^ and skeletal muscle radiodensity <41 HU or BMI >25 kg/m^2^ and skeletal muscle radiodensity <33 HU

**Table 2 cancers-12-01187-t002:** Clinicopathological characteristics of patients who met the inclusion criteria (n = 523).

Characteristic		n = 523 (%)
	Clinico-pathological	
**Age**	<65	226 (43.2)
	65–74	165 (31.5)
	>74	132 (22.5)
**Sex**	Male	266 (50.9)
	Female	257 (49.1)
**Cancer Location**	Lung	177 (33.8)
	Gastrointestinal	180 (34.4)
	Other	166 (31.7)
**Metastatic Disease**	No	88 (16.8)
	Yes	435 (83.2)
	Previous Anti-Cancer Therapy	
**Chemotherapy**	No	149 (28.5)
	Yes	374 (71.5)
**Radiotherapy**	No	362 (69.2)
	Yes	161 (30.8)
**Hormones**	No	470 (89.9)
	Yes	53 (10.1)
	Performance status	
**ECOG-PS**		
**Low Risk**	0/1	255 (48.8)
**Intermediate Risk**	2	204 (39.0)
**High Risk**	3/4	64 (12.2)
**Timed up and go test ˥**	Pass	125 (30.9)
	Fail	279 (69.1)
**Two-minute walk test ˦**	Pass	10 (2.5)
	Fail	393 (97.5)
**Hand grip strength test ˧**	Pass	74 (62.2)
	Fail	45 (37.8)
	Systemic Inflammation	
**mGPS**		
**Low Risk**	0	217 (41.5)
**Intermediate Risk**	1	91 (17.4)
**High Risk**	2	215 (41.1)
	Body composition	
**BMI**	≤20.0 kg/m^2^	74 (14.1)
	20–21.9 kg/m^2^	70 (13.4)
	22–24.9 kg/m^2^	117 (22.4)
	25–27.9 kg/m^2^	107 (20.5)
	≥28.0 kg/m^2^	155 (29.6)
**% Weight Loss**	<2.5	292 (56.0)
	≥2.5	231 (44.0)
**BMI/WL grade**		
**Low Risk**	0/1	276 (52.8)
**Intermediate Risk**	2/3	178 (34.0)
**High Risk**	4	69 (13.2)
**Subcutaneous fat index ˨**	Low	54 (28.1)
	High	138 (71.9)
**Visceral obesity ˨**	Low	79 (41.1)
	High	113 (58.9)
**Low skeletal muscle index ˩**	No	162 (53.3)
	Yes	142 (46.7)
**Low skeletal muscle radiodensity ˪**	No	116 (39.7)
	Yes	176 (60.3)

˥: 404, ˦: 403, ˧: 119, ˨: 192, ˩: 304, ˪: 292.

**Table 3 cancers-12-01187-t003:** The relationship between Eastern Cooperative Oncology Group-Performance status (ECOG-PS) and measures of body composition and physical function in patients with advanced cancer (n = 523).

**High subcutaneous fat index n = 192**	**ECOG-PS 0/1**	**ECOG-PS 2**	**ECOG-PS 3/4**	**All**	***p***
**No**	28 (30.8)	19 (24.7)	7 (29.2)	54 (28.1)	0.677
**Yes**	63 (69.2)	58 (75.3)	17 (70.8)	138 (71.9)	
**All**	91	77	24	192	
**High visceral obesity n = 192**	**ECOG-PS 0/1**	**ECOG-PS 2**	**ECOG-PS 3/4**	**All**	***p***
**No**	38 (41.8)	33 (42.9)	8 (33.3)	79 (41.1)	0.700
**Yes**	53 (58.2)	44 (57.1)	16 (66.7)	113 (58.9)	
**All**	91	77	24	192	
**Low skeletal muscle index n = 304**	**ECOG-PS 0/1**	**ECOG-PS 2**	**ECOG-PS 3/4**	**All**	***p***
**No**	101 (59.8)	49 (45.8)	12 (42.9)	162 (53.3)	0.039
**Yes**	68 (40.2)	58 (54.2)	16 (57.1)	142 (46.7)	
**All**	169	107	28	304	
**Low skeletal muscle radiodensity n = 292**	**ECOG-PS 0/1**	**ECOG-PS 2**	**ECOG-PS 3/4**	**All**	***p***
**No**	74 (46.5)	40 (38.5)	2 (7.4)	116 (39.7)	0.001
**Yes**	85 (53.5)	66 (61.5)	25 (92.6)	176 (60.3)	
**All**	159	104	27	292	
**Timed up and go test failure n = 404**	**ECOG-PS 0/1**	**ECOG-PS 2**	**ECOG-PS 3/4**	**All**	***p***
**No**	94 (54.3) \	29 (17.0)	2 (3.3)	125 (30.9)	<0.001
**Yes**	79 (45.7)	142 (83.0)	58 (96.7)	279 (69.1)	
**All**	173	171	60	404	
**Hand grip strength test failure n = 119**	**ECOG-PS 0/1**	**ECOG-PS 2**	**ECOG-PS 3/4**	**All**	***p***
**No**	56 (68.3)	16 (48.5)	2 (50.0)	74 (62.2)	0.123
**Yes**	26 (31.7)	17 (51.5)	2 (50.0)	45 (37.8)	
**All**	82	33	4	119	

**Table 4 cancers-12-01187-t004:** The relationship between modified Glasgow Prognostic Score (mGPS), and measures of body composition and physical function in patients with advanced cancer (n = 523).

**High subcutaneous fat index n = 192**	**mGPS = 0**	**mGPS = 1**	**mGPS = 2**	**All**	***p***
**No**	22 (29.3)	5 (16.7)	27 (31.0)	54 (28.1)	0.306
**Yes**	53 (70.7)	25 (83.3)	60 (69.0)	138 (71.9)	
**All**	75	30	87	192	
**High visceral obesity n = 192**	**mGPS = 0**	**mGPS = 1**	**mGPS = 2**	**All**	***p***
**No**	32 (42.7)	9 (30.0)	38 (43.7)	79 (41.1)	0.398
**Yes**	43 (57.3)	21 (70.0)	49 (56.3)	113 (58.9)	
**All**	75	30	87	192	
**Low skeletal muscle index n = 304**	**mGPS = 0**	**mGPS = 1**	**mGPS = 2**	**All**	***p***
**No**	72 (55.4)	27 (61.4)	63 (48.5)	162 (53.3)	0.273
**Yes**	58 (44.6)	17 (38.6)	67 (51.5)	142 (46.7)	
**All**	130	44	130	304	
**Low skeletal muscle radiodensity n = 292**	**mGPS = 0**	**mGPS = 1**	**mGPS = 2**	**All**	***p***
**No**	62 (50.4)	15 (34.1)	39 (31.2)	116 (39.7)	0.006
**Yes**	61 (49.6)	29 (65.9)	86 (68.8)	176 (60.3)	
**All**	123	44	125	292	
**Timed up and go test failure n = 404**	**mGPS = 0**	**mGPS = 1**	**mGPS = 2**	**All**	***p***
**No**	66 (41.3)	21 (27.6)	38 (22.6)	125 (30.9)	0.001
**Yes**	94 (58.8)	55 (72.4)	130 (77.4)	279 (69.1)	
**All**	160	76	168	404	
**Hand grip strength test failure n = 119**	**mGPS = 0**	**mGPS =1**	**mGPS = 2**	**All**	***p***
**No**	44 (77.2)	8 (53.3)	22 (46.8)	74 (62.2)	0.005
**Yes**	13 (22.8)	7 (46.7)	25 (53.2)	45 (37.8)	
**All**	57	15	47	119	

**Table 5 cancers-12-01187-t005:** The relationship between body mass index/weight loss (BMI/WL) grade and measures of body composition and physical function measurements in patients with advanced cancer (n = 523).

**High subcutaneous fat index n = 192**	**BMI/WL grade 0/1**	**BMI/WL grade 2/3**	**BMI/WL grade 4**	**All**	***p***
**No**	20 (22.0)	23 (30.7)	11 (42.3)	54 (28.1)	0.104
**Yes**	71 (78.0)	52 (69.3)	15 (57.7)	138 (71.9)	
**All**	91	75	26	192	
**High visceral obesity n = 192**	**BMI/WL grade 0/1**	**BMI/WL grade 2/3**	**BMI/WL grade 4**	**All**	***p***
**No**	30 (33.0)	33 (44.0)	16 (61.5)	79 (41.1)	0.027
**Yes**	61 (67.0)	42 (56.0)	10 (38.5)	113 (58.9)	
**All**	91	75	26	192	
**Low skeletal muscle index n = 304**	**BMI/WL grade 0/1**	**BMI/WL grade 2/3**	**BMI/WL grade 4**	**All**	***p***
**No**	93 (57.8)	56 (51.9)	13 (37.1)	162 (53.3)	0.080
**Yes**	68 (42.2)	52 (48.1)	22 (62.9)	142 (46.7)	
**All**	161	108	35	304	
**Low skeletal muscle radiodensity n = 292**	**BMI/WL grade 0/1**	**BMI/WL grade 2/3**	**BMI/WL grade 4**	**All**	***p***
**No**	70 (45.8)	41 (39.4)	5 (14.3)	116 (39.7)	0.003
**Yes**	83 (54.2)	63 (60.6)	30 (85.7)	176 (60.3)	
**All**	153	104	35	292	
**Timed up and go test failure n = 404**	**BMI/WL grade 0/1**	**BMI/WL grade 2/3**	**BMI/WL grade 4**	**All**	***p***
**No**	68 (33.7)	41 (28.9)	16 (26.7)	125 (30.9)	0.473
**Yes**	134 (66.3)	101 (71.1)	44 (73.3)	279 (69.1)	
**All**	202	142	60	404	
**Hand grip strength test failure n = 119**	**BMI/WL grade 0/1**	**BMI/WL grade 2/3**	**BMI/WL grade 4**	**All**	***p***
**No**	47 (63.5)	21 (58.3)	6 (66.7)	74 (62.2)	0.835
**Yes**	27 (36.5)	15 (41.7)	3 (33.3)	45 (37.8)	
**All**	74	36	9	119	

**Table 6 cancers-12-01187-t006:** The relationship between ECOG-PS, mGPS, BMI/WL grade and skeletal muscle index, skeletal muscle radiodensity and physical function in patients with advanced cancer (n = 523).

**High subcutaneous fat index**	**Univariate**	***p*** **-value**	**Multivariate**	***p*** **-value**
**ECOG-PS**	1.13 (0.71–1.78)	0.617	─	0.319
**mGPS**	0.95 (0.67–1.34)	0.776	─	0.995
**BMI/WL Grade**	0.62 (0.40–0.97)	0.036	0.62 (0.40–0.97)	0.036
**High visceral obesity**	**Univariate**	***p*** **-value**	**Multivariate**	***p*** **-value**
**ECOG-PS**	1.12 (0.74–1.70)	0.606	─	0.254
**mGPS**	0.97 (0.71–1.33)	0.865	─	0.844
**BMI/WL Grade**	0.57 (0.38–0.87)	0.009	0.57 (0.38–0.87)	0.009
**Low skeletal muscle index**	**Univariate**	***p*** **-value**	**Multivariate**	***p*** **-value**
**ECOG-PS**	1.53 (1.08–2.17)	0.016	1.90 (1.51–2.39)	<0.001
**mGPS**	1.15 (0.90–1.47)	0.264	─	0.768
**BMI/WL Grade**	1.44 (1.03–2.00)	0.033	─	0.106
**Low skeletal muscle radiodensity**	**Univariate**	***p*** **-value**	**Multivariate**	***p*** **-value**
**ECOG-PS**	2.01 (1.36–2.98)	<0.001	1.68 (1.11–2.55)	0.013
**mGPS**	1.50 (1.16–1.95)	0.002	1.32 (1.01–1.73)	0.049
**BMI/WL Grade**	1.77 (1.23–2.55)	0.002	1.50 (1.02–2.19)	0.037
**Timed up and go test failure**	**Univariate**	***p*** **-value**	**Multivariate**	***p*** **-value**
**ECOG-PS**	5.84 (3.79–9.00)	<0.001	5.84 (3.79–9.00)	<0.001
**mGPS**	1.56 (1.22–1.98)	<0.001	─	0.231
**BMI/WL Grade**	1.20 (0.89–1.61)	0.232	─	0.484
**Hand grip strength test failure**	**Univariate**	***p*** **-value**	**Multivariate**	***p*** **-value**
**ECOG-PS**	1.93 (0.97–3.84)	0.060	─	0.213
**mGPS**	1.95 (1.29–2.97)	0.002	1.95 (1.29–2.97)	0.002
**BMI/WL Grade**	1.05 (0.59–1.89)	0.862	─	0.621

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
