# Peer review of "The Relationship between ECOG-PS, mGPS, BMI/WL Grade and Body Composition and Physical Function in Patients with Advanced Cancer"

_cancers, 2020, doi:10.3390/cancers12051187_

Round 1

Reviewer 1 Report

Throughout – AS a reader, I am less inclined to like abbreviations that are not common (ie units of measurement). I would limit the number of abbreviations to less than 5 in any manuscript and even more so when they are specific to a particular test, such as body composition in this case. For example, SMI, SFI, SFA, SMA, TFI, VFI, TUG are not common and although used frequently throughout the manuscript it would not added significantly to length if each abbreviation is spelt out in full. It would aid the reading flow of the manuscript for non-specialist readers.

Abstract –

Line 25 & 33 – abbreviations need to explained. Appreciate they are spelt out in the introduction or methods, but difficult for readers to know whether to read full article when abstract is full of uncommon abbreviations. It would be preferable to allow a slightly longer abstract without any abbreviations to aid flow of the study summary.

Line 35 – could the authors clarify in the abstract the direction of the association – e.g high PS and high mGPS are associated with measures of reduced body composition and poorer physical function.

Methods

Line 72 – please clarify whether cancers were of any specific histological subtype or where all cancers included in analysis regardless of the cancer subtype. This should be made clear in the manuscript.

Line 76 – where all cancer patients at the same treatment stage – first line, second line etc? Where the patients analysed at the beginning of treatment and only these readings were included in analysis or did it not matter for analysis? This should be clear for readers and future researchers comparing their own studies with this dataset.

Line 79 – Where readings for all of the prognostic markers within 3 months of starting the study similar to the measurements of body composition? This should be made clear in the manuscript.

Table 1 – please include a note under the table to contain abbreviations. These may be known to researchers within the field, but uncommon to new researchers and they need to have easy access to remember the abbreviations.

Line 117 – space between and    handgrip needs fixing

Line 118 – what was the period of weight loss associated with – last 3 or 6 months?

Line 126 – Statistical methods –

More detail about what method of multivariate analysis was used – forward, backward, what was the cut-off for inclusion in each step.

Also could the authors please adjust or stratify the results for the major subpopulations? The lung/CRC populations are quite large (n~ 180 patients each) and are a large enough cohort to stratify the results to understand if associations are cancer-type specific.

Discussion

Line 248 – could the authors discuss the limitations of study based on the generalisability of the results to the wider cancer population when the study cohort was very heterogenous – e.g. mixed cancer types, different stages of cancer, different stages of prior and current treatment. How would the authors propose to fix this with future studies?

Line 254– could the authors clarify in the abstract the direction of the association – e.g high PS and high mGPS are associated with measures of reduced body composition and poorer physical function.

Reviewer 2 Report

The authors did a further analysis an an existing database to study the value of certain scoring systems and measures of the subjects. The study is interesting, but there need to be some clarifications.

I find the paper to have too many abbreviations that limit the reading substantially. 

69: Were there already data published from this database? Please cites those

83-89: Why did you chose to put the data into your analysis in categories, while the data are continuous? I also do not understand the focus on fail of a test. This seems to be very artificial.

103: Why is SMI named SMI (martin), and not just SMI and indicated in the text the source of your categories?

Table 2 and other tables

If only 10 out of 403 subjects pass the 2MWT, how can it be a factor in the other tables?

Round 2

Reviewer 2 Report

The authors response is sufficient.